# BK Polyomavirus Nephropathy in Kidney Transplantation: Balancing Rejection and Infection

**DOI:** 10.3390/v13030487

**Published:** 2021-03-16

**Authors:** Chia-Lin Shen, Bo-Sheng Wu, Tse-Jen Lien, An-Hang Yang, Chih-Yu Yang

**Affiliations:** 1Division of Nephrology, Department of Medicine, Taipei Veterans General Hospital, Taipei 11217, Taiwan; annieshen1982@gmail.com (C.-L.S.); jenmei2005@gmail.com (T.-J.L.); 2Jen Chia Clinic, New Taipei City 24147, Taiwan; 3School of Medicine, Faculty of Medicine, National Yang Ming Chiao Tung University, Taipei 11221, Taiwan; boshengw16507@gm.ym.edu.tw (B.-S.W.); ahyang@vghtpe.gov.tw (A.-H.Y.); 4School of Medicine, Institute of Clinical Medicine, National Yang Ming Chiao Tung University, Taipei 11221, Taiwan; 5Jen Mei Clinic, New Taipei City 24759, Taiwan; 6Department of Pathology, Taipei Veteran General Hospital, Taipei 11217, Taiwan; 7Stem Cell Research Center, National Yang Ming Chiao Tung University, Taipei 11221, Taiwan; 8Center for Intelligent Drug Systems and Smart Bio-Devices (IDS2B), Hsinchu 30010, Taiwan

**Keywords:** BK polyomavirus nephropathy, kidney transplant, acute rejection, immunosuppressants, tacrolimus

## Abstract

BK polyomavirus nephropathy (BKVN) and allograft rejection are two closely-associated diseases on opposite ends of the immune scale in kidney transplant recipients. The principle of balancing the immune system remains the mainstay of therapeutic strategy. While patient outcomes can be improved through screening, risk factors identification, and rapid reduction of immunosuppressants, a lack of standard curative therapy is the primary concern during clinical practice. Additionally, difficulty in pathological differential diagnosis and clinicopathology’s dissociation pose problems for a definite diagnosis. This article discusses the delicate evaluation needed to optimize immunosuppression and reviews recent advances in molecular diagnosis and immunological therapy for BKVN patients. New biomarkers for BKVN diagnosis are under development. For example, measurement of virus-specific T cell level may play a role in steering immunosuppressants. The development of cellular therapy may provide prevention, even a cure, for BKVN, a complex post-transplant complication.

## 1. Introduction

BK polyomavirus nephropathy (BKVN) and allograft rejection are two significant post-transplant complications on opposite ends of the immune spectrum (Figure 1). Parajuli et al. studied 3-year outcomes between these two diseases retrospectively. While BKVN and rejection are both prominent causes of kidney damage, renal function 3 years after diagnosis was worse for BKVN than for rejection [1]. The leading cause of BKVN is over-immunosuppression that reactivated the latent BK polyomavirus (BKPyV) within the recipient or reinforced BKPyV infection inside the allograft. No effective direct antiviral therapy is currently available; thus, since the first case was identified in 1971, immunosuppressant (IS) reduction remains the primary strategy for BKVN [2]. On the other hand, insufficient IS usage predisposes acute or chronic rejection, leading to graft function decline or graft loss as well.

Early diagnosis based on onset time and clinical manifestation is difficult due to similar clinical presentation of graft rejection and BKVN. Therefore, the highest principle in clinical practice is keeping a balance between rejection and infection [3]. This article discusses the evaluations needed for optimal immunosuppression to avoid infection or reactivation of the BKPyV in kidney transplant recipients (KTRs). In the case of confirmed BKPyV infection, control of the disease progression to preserve the graft function is also reviewed.

## 2. About the BKPyV

BKPyV is a highly prevalent polyomavirus specific to the human host [4]. As a double-stranded DNA virus, its genome consists of the early coding region, late coding region, and a non-coding control region (NCCR) in between [5]. The early region usually codes for the replication proteins, including the small tumor antigens, the large tumor antigens (TAgs), and agnoprotein. The late region codes for structural proteins VP1, VP2, and VP3 [6]. The microRNAs expression was transcribed from the 3′ end of the TAgs and act as a regulator in BKPyV infection [7]. The NCCR contains the genome of promoters of the early and late regions, transcriptional start sites, and the origin of replication. It also provides binding sites for host cellular regulatory factors. NCCR variation exists between BKPyV isolates, and the rearranged forms of NCCR are associated with disease [8]. The high heterogeneity of NCCR allows for environmental adaptation and higher pathogenicity for disease progression [6].

Cellular immunity is critical for the immune response during BKPyV viremia and BKVN. Innate immune response serves as the first line of defense against the primary infection [9]. Dendritic cells are critical in the induction of adaptive immune response [10]. Womer et al. reported that the number of peripheral blood dendritic cells is lower in KTRs developed BKVN. They also revealed that KTRs with fewer dendritic cells before transplantation are more likely to be associated with BKVN [11]. Furthermore, BKPyV can decrease the natural killer cell-mediated cytotoxicity by inhibiting the identification of natural killer cells [12]. Other innate immune mediators are associated with renal inflammation [13]. Adaptive immune response develops after exposure to viral antigens. Humoral response works via neutralizing antibodies to defend the further viral infectious process. Studies showed seronegative recipients have higher risks in viremia and subsequent BKVN than seropositive recipients as humoral immunity may help limit BKPyV infection [14,15,16,17,18,19]. Meanwhile, recipients paired with seropositive donors have a higher post-transplant BK-specific-antibody titer than the seronegative donor group [20]. It means that BKPyV infection from the donor can induce the humoral immune response [21]. However, the virus can hide away from neutralization with a mutation in viral antibody receptors [22,23]. In this situation, latent viral reactivation can be well-controlled by antiviral memory T cells [4]. Cellular immunity offers more effective infection control because of pathogen detection and cytotoxicity [21]. Both CD4+ and CD8+ T cells are important, especially the polyfunctional BKPyV-specific T cells [24,25]. After kidney transplant, KTRs with viruria but no viremia have positive BKPyV-specific T cell response [26]. Conversely, there is no BKPyV-specific T cell response in KTRs with BKPyV viremia or BKVN [26,27,28,29,30]. Also, quick BK-specific T cell response was noted in the viremia-resolved group, while the response was only noted after reduced IS in the developed BKVN group [28,31]. These studies concluded that it is crucial for KTRs to reconstitute the BKPyV-specific T cells to fight against BKPyV infection.

During the first decade of childhood, the primary exposure to BKPyV, often with subclinical symptoms, resulted in 80–90% of adults developed antibodies against BKPyV [32,33]. The natural transmission route is still unknown [34]. After the primary infection, the virus remains latent in the kidney, peripheral-blood leukocytes, and possibly the brain [35]. The viral reactivation occurs while the host immunity is over-suppressed, resulting in viral replication with consequent tubular cell lysis and viruria. BKPyV replication ensues in the renal interstitium, leading to the destruction of the tubular capillary wall subsequently cross into the blood, causing viremia. Viral invasion of tissue progressively cause cell necrosis and tissue inflammation [36]. BKPyV reactivation presented as viremia usually happens in the first month post-transplant in KTRs. The incidence peaks around 28–31% at month 3 and month 12 after kidney transplantation, with cases rarely seen at month 18 [37,38]. In the KTR population, the incidence of BKPyV viruria is 30–40%, BKPyV viremia is 13%, and BKVN is 8% [39]. High-level BKPyV viruria progress to viremia after a median of 4 weeks, and approximately a median of 8 weeks later, viremia may lead to BKVN [40,41]. The clinical presentation of BKPyV infection may range from asymptomatic to progressive renal function decline, and others are incidental findings at protocol allograft biopsy [42]. The laboratory clues may be ranged from normal results to elevated serum creatinine, mild proteinuria (48%), or hematuria (19%) [43]. Without screening and treatment, the natural course of BKVN leads to 50% graft loss [44,45].

## 3. Screening and Diagnosis

Early diagnosis of BKVN usually results in better allograft survival than the advanced disease [43,46]. Due to limited treatment options, screening for BKPyV replication is recommended to avoid further kidney histologic involvement. Intensive screening by measuring blood BKPyV DNA can help patients at risk of BKVN preserve allograft function [47,48]. Monitoring of disease progression can be done through urine or blood polymerase chain reaction (PCR). The threshold value of urine viral load is 1 × 10^7^ copies/mL. Viruria has a negative predictive value of 100% for BKVN, a positive predictive value of 31–67%, a sensitivity of 100%, and a specificity of 92–96% [48]. The threshold value of blood PCR is 1 × 10^4^ copies/mL. Viremia has a negative predictive value of 100% for BKVN, a positive predictive value of 50–82%, a sensitivity of 100%, and a specificity of 88–96% [44,49]. The higher positive predictive value of viremia over viruria explains the 2019 Guidelines from the American Society of Transplantation Infectious Diseases Community of Practice (AST-IDCOP), which suggested all KTRs should be screened for blood BKPyV DNA monthly until month 9 and then every 3 months until 2 years post-transplant [50]. Decoy cells, infected tubular epithelial cells identified by the urine cytology examination, are also standard screening methods but wholly depend on pathologists’ experience [49]. A Japanese study showed an increasing trend of decoy cells in the BK viremia group and suggested decoy cells can predict early BKPyV infection with continuous and careful monitoring [51]. Additionally, the 2009 KDIGO guideline indicated that in the case of unexplained allograft dysfunction or recent IS dosage increases, one should be cautious of BKPyV [52].

The diagnosis of BKVN relies on clinical judgment and pathological morphologic diagnosis [43]. Presumptive nephropathy, meaning a primary diagnosis without histologic confirmation, is defined as plasma BK viral DNA PCR load >10,000 copies/mL with urinary viral shedding for more than 2 weeks with or without renal function decline [53]. However, once suspected of renal function decline or possible acute rejection, renal biopsy should still be performed before reducing IS dosage [50]. Morphological diagnosis by light microscopy is limited due to similarities between early BKVN and other diagnoses such as acute rejection or calcineurin inhibitor (CNI) toxicity. Definite diagnosis of BKVN can be achieved through a cytopathic change of tubular epithelial cells combined with in situ hybridization against SV40 or Tag [54]. A unified diagnostic criterion is crucial for the comparability of different studies. However, previous morphology diagnosis classification is yet to provide statistical discriminative power for the clinical correlation sufficient enough to revise the classification [55]. AST-IDCOP revised the histological classification with a more detailed description of the degree of interstitial inflammation and the area of the biopsy tissue in 2013 [56]. Banff 2017 working group enrolled multicenter retrospective study analyzed confirmed BKVN systematically to develop a morphologic classification. Intrarenal BKPyV viral load and the Banff interstitial cortical fibrosis score are two independent factors with a significant correlation with clinical presentation and graft outcome [43]. AST-IDCOP 2019 recommended that histological findings of proven BKVN be reported based on AST-IDCOP 2013 and the Banff 2017 classification [50]. As for cases with coexisting BKVN and acute rejection, tubulitis and peritubular inflammation examination by immunohistochemistry and electron microscopy should be performed. The presence of endarteritis, fibrinoid vascular necrosis, glomerulitis, or C4d deposits along peritubular capillaries should be documented for the diagnosis of coexisting BKVN and acute rejection [57,58,59].

## 4. Balancing the Rejection and Infection

BKPyV reactivation is induced by relative or absolute immunodeficient status, such as pregnancy, cancer, HIV infection, and diabetes [60]. Typical BKPyV reactivation occurs early after transplantation or after over immunosuppression [61]. BKPyV infection or reactivation can be managed by balancing the immune system. In other words, IS dose should be delicately reduced to avoid allograft rejection. In this part, we discuss methods to reduce the possibility of infection or reactivation in addition to the management strategies of BKPyV infection.

### 4.1. Risk Factors for BKPyV Infection or Reactivation

Risk factor identification for BKPyV is essential. The studied risk factors for BKPyV infection can be assorted into several categories: Donor risk factors, recipients risk factors, and transplant risk factors (Figure 2) [16,18,39,56,62,63,64,65,66,67,68,69,70,71,72]. A systemic review revealed the most relevant risk factors for BKPyV viremia after kidney transplantation were a tacrolimus regimen, a deceased donor, a male recipient, a history of the previous transplant, age at transplantation, ureteral stent use, delayed graft function, and acute rejection episodes [73]. Due to the low frequency of the BKVN, the sample size of each study is small; therefore, it is difficult to reach statistically significant results in BKVN studies. Prince et al. enrolled 34 BKVN patients in a single-center and reported that tacrolimus, mycophenolate mofetil (MMF), and acute rejection were significant risk factors for BKVN [74]. Pai et al. published another single-center retrospective study, where 14 BKVN patients were assessed for associated risk factors of BKVN. Episodes of rejection, transplantation of >1 organ, positive *cytomegalovirus* (CMV) serology in both donor and recipient, and a more significant cumulative dose of daclizumab use at the time of induction were statistically significant risk factors for the development of BKVN [69]. Prince et al. suggested that BKVN only manifests while the host immunity is over-suppressed, whereas acute rejection independently plays a role regardless of therapeutic regimens [74]. Hence, understanding pre- and post-transplant risk factors can be helpful to balance the infection and rejection.

Tacrolimus itself is a potent IS compared with cyclosporine with less acute rejection rate, as evident by a phase III multicenter trial [75]. A meta-analysis showed less graft loss, less acute rejection rate, and less steroid-resistant rejection when compared with cyclosporine [76]. Among all the IS, emerging data suggest that tacrolimus use possesses the greatest risk for BKVN [77]. Hirsch et al. analyzed the DIRECT trial, which compared tacrolimus to cyclosporine in a combined regimen prospectively. A higher incidence rate of BK viremia in the tacrolimus group 6 months after transplant was reported [47]. Benavides et al. [77] and Moscarelli et al. [78] both found that mammalian target of rapamycin (mTOR) inhibitor is less likely to be associated with BK viremia and BKVN. Hirsch et al. reported mTOR inhibitor sirolimus could inhibit BKPyV replication during gene expression while tacrolimus plays a role in activating replication via FK binding protein-12 kDa [79]. This study provides rationales for the further clinical trial of anti-BKPyV therapy. Ureter stent use is another crucial transplant risk factor not related to immune status, especially for postoperative recovery. The association between ureteral stents and BKPyV is well documented because tubular and urothelial cell injury allows for BKPyV replication [70,71].

Both BKPyV serostatus of donor and recipient are important risk factors. Wunderink et al. published the largest research to date showing that donor seropositivity was strongly associated with the occurrence of recipient viremia and BKVN (*p* < 0.001, Student’s t-test). The results also pointed out that when high-BKPyV-seroreactive donors are paired with low-seroreactive recipients, the recipients have a 10-fold increased risk of BKPyV viremia [16]. BKPyV serostatus can be used as a method for risk stratification for BKPyV reactivation. Sood et al. showed that viremia is the highest in the donor-seropositive-recipient-positive group but is the lowest in the donor-seronegative-recipient-seronegative group [18]. These studies show strong evidence for donor-origin BKPyV infection as a vital transmission source. Several other studies reached the same conclusion through ELISA or neutralization inhibition assays [15,17,19,80,81,82]. An additional clinical guideline may be needed to define serostatus, cut-off values, and standard testing methods for clinical use.

### 4.2. Tacrolimus Dosage Monitoring

Tacrolimus, a potent immunosuppressor, decreases the incidence rate of early acute rejection and graft failure rate [82,83]. Tacrolimus drug-level monitoring is important in post-transplant care because of the pharmacological properties: Non-linear concentration-effect relationship, narrow therapeutic window, and nephrotoxicity [84]. Other related issues include metabolic side effects, CNI nephrotoxicity, and over-suppression-related opportunistic infection. Moreover, inter-patient variability and intra-patient variability may cause fluctuation of blood tacrolimus level even with stable prescription dosage [85]. Outcome-association between the drug-level and graft function is evident [86,87,88,89]: Insufficient tacrolimus may increase rejection risk while overdosing brings possible toxicity and infection [90]. Tacrolimus trough level >10 ng/mL is associated with BKPyV infection after kidney transplantation [66]. Due to the reasons stated above, an optimal method to tailor IS dosage with proper monitoring is a topic worth exploration.

Trough-level monitoring of tacrolimus is the standard of care in clinical practice. The result of a clinical phase III trial indicated a trough level between 10–15 ng/mL during the 3-month-period after transplantation and 5–10 ng/mL afterward [91]. A meta-analysis of 10 studies and almost 6000 individuals showed maintaining tacrolimus blood concentration at 5–9.5 ng/mL within the first year may be the most effective method to prevent acute rejection [92]. Steroids are cytochrome P3A (CYP3A) inducer, and tacrolimus is involved in the cytochrome P450 catalytic cycle. Tacrolimus level may increase when high dose steroid is used. Once the steroid is tapered, tacrolimus may become overdosed with consequent nephrotoxicity [93]. The Symphony study minimized tacrolimus dose and observed that a lower tacrolimus level (3–7 ng/mL) combined with daclizumab induction, MMF, and steroids, possessed the lowest risk of acute rejection at month 12 post-transplant. This suggested that tacrolimus blood level can be lower than usual when combined with an mTOR inhibitor in the maintenance stage [94]. Difficult dosage titration in a complex clinical situation should be managed by an experienced kidney transplant specialist in a high-volume medical center.

#### 4.2.1. Inter-Patient Variability of Blood Tacrolimus Level

Tailored usage of tacrolimus requires delicate clinical adjustments based on drug-level monitoring due to the inter-patient variability. The inter-patient blood tacrolimus level varies because of individual-specific pharmacokinetics and pharmacogenetics. Factors affecting inter-patient heterogeneity include age, race, hepatic dysfunction, blood albumin, hematocrit, and diurnal rhythm [95]. Higher tacrolimus doses are needed for pediatric patients to reach the same trough level, likely due to the higher CYP3A gene expression in children [96,97]. African-American recipients need higher tacrolimus doses due to reduced bioavailability, suggesting racial differences in intestinal CYP3A or P-glycoprotein activity [98,99,100]. Because tacrolimus strongly binds to red blood cells and albumin, its trough level seems to be lower after the transplant but is then elevated after patient recovery [101]. As for the diurnal rhythm, the higher daytime clearance is probably due to the circadian effect on gastric emptying time and intestinal perfusion for drug absorption [102,103].

#### 4.2.2. Intra-Patient Variability of Blood Tacrolimus Level

Generally, intra-patient variability (IPV) is the fluctuation in tacrolimus trough concentrations of a single individual with unchanged tacrolimus dose over a period of time. On average, the tacrolimus IPV is between 15% and 30%, while others reported a wider IPV range from lower than 5% to over 50% [85]. Several determinants may contribute to IPV; the impact from high to low include medication nonadherence, drug-drug interaction, nutritional interferences, concurrent disease, analytical assay, genetics, and generic tacrolimus substitution [104]. Nonadherence to the IS drug is the most common and the main determinant factor of high tacrolimus IPV. A meta-analysis pointed out a 7-fold risk of graft failure between nonadherent and adherent groups [105]. Macrolide antibiotics, azole antifungals, rifampin, glucocorticoids, calcium channel blockers, and anti-epileptic agents may influence the tacrolimus IPV by altering the CYP3A activities. Patients should avoid over-the-counter drugs, grapefruit, pomelo, high-fat meal, and concomitant food ingestion [106,107,108,109,110]. Illnesses like diarrhea, anemia, hypoalbuminemia, and hyperlipidemia are related clinical problems [111]. When no obvious interference factor is found, the genetic difference might explain the fluctuation [112].

Multiple large studies have demonstrated significant negative consequences of the high tacrolimus IPV, including graft survival, acute rejection, de novo donor-specific antibody (dnDSA), and chronic immunologic-mediated graft injury. The first long-term outcome study by Borra et al. reported that high tacrolimus IPV was related to 1-year post-transplant graft function decline [113]. Shuker et al. enrolled a larger cohort study with primary endpoints, including graft failure, late biopsy-proven acute rejection, transplant glomerulopathy, or doubling time **of** serum creatinine concentration. The result revealed that a high tacrolimus IPV was an independent predictor of inferior graft outcomes [114]. Likewise, Rozen-Zvi et al. showed a higher mean tacrolimus IPV (mean IPV 34.8 ± 21.3%) was associated with worse graft survival within the 6-month post-transplant phase [115]. Ro et al. were the first to present an association between high tacrolimus IPV and acute rejection risk [116]. In our study, high tacrolimus IPV was associated with coexist acute rejection and BKVN [67].

Further research may aim to quantify IPV as a scale for surveillance and reduce acute rejection incidence [117]. Meanwhile, evidence was connected to tacrolimus IPV with dnDSAs and antibody-mediated rejection. A Spanish study with 6.6 years of follow-up was designed to evaluate the incidence of dnDSAs and graft survival in relation to tacrolimus IPV. The primary endpoint showed dnDSA development was associated with worse graft survival (*p* < 0.001), while tacrolimus IPV was associated with dnDSA development (*p* = 0.002). The secondary endpoint showed high tacrolimus IPV associated with an increased risk of graft loss. To sum it up, tacrolimus IPV is an independent factor for graft loss and a strong risk factor for dnDSA development [118]. Sablik et al. found that recipients with chronic active antibody-mediated rejection showed a significant association with lower allograft survival in the high tacrolimus IPV group [119]. Vanhove et al. designed a study according to protocol biopsy and IPV tertiles to provide a direct and robust histology link to clinical correlation. They proved a positive association between tacrolimus IPV and the evolution of acute on chronic histologic lesions. It suggested that tacrolimus IPV monitoring can predict chronic histologic lesions’ progression before the onset of renal dysfunction [120]. Gradually it became clear that high tacrolimus IPV is related to acute and chronic rejection, dnDSA formation in high immunologic risk recipients, histological fibrotic change, and poor allograft outcome.

The current clinical focus of tacrolimus IPV is the reduction of IPV, a standardized formula to calculate tacrolimus IPV, and whether decreasing IPV improves outcome or not. As high tacrolimus IPV can cause coexistent acute rejection and BKVN, this clinical application matches our topic. Reduction of IPV in practice may include careful drug usage with food or other drugs and regular drug usage without skipping doses. Previous phase III and IV studies in de novo initiation confirmed the same medicinal effect of once-daily and twice-daily tacrolimus in preventing acute rejection and graft loss [121,122,123,124]. Some Investigators focused on switching twice-daily tacrolimus to a once-daily formulation for better adherence, and the result favored the once-daily extended-release tacrolimus [125,126]. The first prospective randomized control trial by McGillicuddy et al. adjusted the nonadherence by electronics significantly decreased tacrolimus IPV [127]. Future studies are warranted for the impact of reducing tacrolimus IPV on graft outcome. Also, a delicate cut-off value of the variability should be identified for better patient risk evaluations and clinical tacrolimus adjustment. In terms of a cut point, patients with tacrolimus IPV >30% or >40% should be considered to be at high risks for BKVN by experts’ opinions [128].

### 4.3. Other Immunosuppressants

The current standard triple regimen tacrolimus-mycophenolate mofetil-steroid is originated from several clinical trials in decades with the advantages of lower rejection rate, or lower required IS dosage. A meta-analysis comparing cyclosporine and tacrolimus reported tacrolimus significantly reduced graft loss and acute rejection rates, but there was no difference in infection between the two groups [76]. In patients with persistent viremia after reducing all the IS, shifting from tacrolimus to cyclosporine is an alternative strategy to alleviate concerns over insufficient immunosuppression [47]. Kim et al. found that high-dose steroid (cumulative intravenous steroids > 2 g within 30 days) may increase BKPyV infection and result in poor long-term graft function [68]. Hirsch et al. found that BKPyV replication within renal tubular epithelial cells is inhibited by sirolimus but is activated by tacrolimus through a pathway involving FKBP-12 [129]. Clinical studies revealed sirolimus-based IS regimen does indeed inhibit BKPyV with a lower incidence rate of BKPyV infection [62,130]. A prospective, controlled study reported that an everolimus-based IS regimen with CNI minimization and MMF discontinuation effectively treated BKVN in KTRs [131]. As for the role of MMF in BKPyV infection, though MMF use has been reported as a risk factor [74], most studies revealed no direct association. Therefore, it remains a controversial topic [132].

### 4.4. Risk Factors for Acute Rejection

Risk factors for acute rejection are important for BKPyV management, especially for the clinical differentiation of renal dysfunction. Pre-transplant donor-specific antibody and HLA mismatch are, respectively, main predictors of antibody-mediated rejection and T cell-mediated rejection [133]. The regimen of IS is the determining factor for the post-transplant risk of acute rejection. Reducing or replacing tacrolimus with an add-on mTOR inhibitor may increase the acute rejection rate [134]. A cohort study with tacrolimus-based triple IS regimen reported mean tacrolimus level < 8 ng/mL in the first year has a strong association of increased DSA development (*p* = 0.005). Mean tacrolimus 4–6 ng/mL may increase acute rejection rate 2.3-fold compared with the 8 ng/mL group [135]. Other risk factors may include younger recipient age, older donor age, African-American ethnicity, delayed onset of graft function, and cold ischemic time over 24 h [52].

### 4.5. Biologic Marker Development in BKVN

While indirect examinations such as creatinine, drug trough level, and urine analysis can provide limited information, the definitive method for BKVN diagnosis is still renal biopsy. Ongoing research and the development of new non-invasive monitoring measurements provide promising biomarkers to assist the definite diagnosis of BKVN. It has been reported that urinary exosomal BK viral microRNA, bkv-miR-B1-5p and bkv-miR-B1-5p/miR-16, have excellent statistical significance for the diagnosis of BKVN, with the area under the curve values of 0.989 and 0.985, respectively [136,137,138]. Dvir et al. hypothesized the association of the interferon-λ family with BKVN due to the antiviral protection of the epithelium. They found a single-nucleotide polymorphism rs12979860 in the genomic region of interleukin-28B has predictive value for identifying high-risk patient progression from viremia to BKVN [139]. Ho et al. described the correlation of urine C-X-C motif chemokine ligand 10 (CXCL10) with BK viremia. The urine CXCL10 represents subclinical inflammation within tubular-interstitial and peritubular capillary spaces in the study [140]. The challenge faced during biomarker development of BKVN is the overlapping pathogenetic mechanisms of BKVN with other allograft injuries, such as rejection and tubular interstitial fibrosis. There are still no mature biomarkers yet and need future research for clinical monitoring and guiding optimal IS adjustment [141].

On the other hand, biomarkers for acute rejection may still be helpful for disease differentiation. Ongoing research for biomarkers intended for the diagnosis, exclusion, or confirmation of acute rejection. Suthanthiran et al. reported a molecular signature of CD3ε mRNA, IP-10 mRNA, and 18S rRNA levels in urinary cells that appear to be diagnostic and prognostic of acute cellular rejection in kidney allografts [142]. Urinary chemokines C-X-C motif chemokine ligand 9 (CXCL9) and CXCL10 are the most well-developed biomarkers for T-cell mediated rejection and acute antibody-mediated rejection [143,144,145,146]. KTRs with low urinary CXCL9 protein levels within the 6-month post-transplant period were less likely to experience future acute rejection between 6 and 24 months (NPV 92.5–99.3%) [144].

Meanwhile, plasma donor-derived cell-free DNA (ddcfDNA) fractions decreased exponentially within 10 days after transplantation to a ddcfDNA threshold value of 0.88% or less. An increase above the reference baseline of 0.88% was associated with acute rejection, but acute pyelonephritis and acute tubular necrosis cannot be excluded [147]. Many other non-invasive ongoing biomarkers are under investigation, and randomized control trials are needed for future implementation into clinical practice.

## 5. Strategies of Immunosuppression Reduction

When a KTR’s immune system is over-suppressed, the incidence of infection increases. Aggressive dosage reduction of IS help immunity recovery, protect allograft outcome, and save lives. Our group studied the timing of the first IS drug reduction of the *Pneumocystis jiroveci* pneumonia survivors and found that a prompt and sufficient reduction of IS dosage significantly improved mortality with minimal risks of in-hospital and long-term acute rejection [148]. Additionally, we also examined KTRs who suffered from severe bacterial pneumonia with respiratory failure and acute kidney injury. Our study showed a minimal risk of acute rejection during 2-year follow-up with a trend, though not significant, improved in-hospital mortality [149]. Both studies reflect that immunosuppression reduction is the right side to choose when facing severe infection in kidney transplants.

According to the KDIGO guideline in 2009, reduction of immunosuppression for BKPyV infection in KTR is still the primary treatment to date (Figure 3) [52]. Any concurrent or increase risk of acute rejection should be taken into consideration. There are no standard regimens, and the outcomes vary between institutes. Currently, there are only meta-analyses and prospective observational studies. A systematic review analyzed 8 cohorts and 13 case-series, showing that a IS reduction alone strategy without additional anti-viral drug use may achieve a relatively low graft loss rate of 0.08 per patient-year in patients with BKVN [150]. Clinically, common stepwise strategies are as follows:A once or twice dose reduction of the CNI by 25–50%, with target tacrolimus trough level < 6 ng/mL followed by reducing the antimetabolite drug by 50%, and lastly discontinuing the latter in the case of high viral load [40].Decrease the antimetabolite drug by 50% or discontinuation, then decrease CNI by 25~50% if viremia does not resolve [48].Reducing both the CNI and the antimetabolite drug simultaneously [151].

Rejection may happen, or donor-specific antibodies may develop due to different responses from individuals. Close monitoring of blood creatinine, plasma BK viral loads, and calcineurin inhibitor levels are required. Donor-specific antibody and blood viral load PCR can be followed up for evaluations [152]. Coexisting acute allograft rejection should be concerned, and renal biopsy should be performed once serum rises with viral load decline.

Recent single-center retrospective studies were followed for a longer duration with a higher case number compared to previous reports. Since no randomized prospective trial has been conducted, the current understanding is based on rationale from large cohort studies. Most of these study designs are separated into two groups: Taper CNI in the first step (CNI first) or taper antimetabolite agent first. Sawinski et al. reported a 3 year-follow-up study showing that decreasing antimetabolite agent first posed no differences in patient and allograft survival rates between patients with and without BK viremia [153]. Seifert et al. also firstly reduced antimetabolite agents in a 10-year cohort and found that the mortality rate was slightly higher in the BK viremia group. However, no differences in rejection rate, death-censored graft survival, and graft function were noted [154]. Bischof et al. retrospectively examined a 6-year cohort and found that the reducing CNI first strategy led to similar long-term outcomes between patients with and without BK viremia, but reducing CNI first posed a low risk for ABMR after viremia clearance [155]. Baek et al. published a 6-year retrospective cohort study that concluded CNI dose reduction by >20% at 1 month after the initial BKPyV detection could increase the risk of acute rejection [156]. Different combination regimens balancing rejection and infection depend on the on-target and off-target properties. CNIs block T cells’ signal transduction that impaired cytokine secretion while mTOR inhibitors or antimetabolite agents do not [157]. Renner et al. showed that the tacrolimus and MMF-based combination increased the risk of BKPyV viremia, which became not different from the cyclosporine/MMF group when tacrolimus was converted to everolimus [24]. As mentioned above, the mTOR inhibitor suppresses BKPyV replication in vitro while tacrolimus activates virus production [79]. Since there is no consensus on using either CNI or MMF reduction strategies, further randomized trials are expected. Schwarz et al. retrospectively studied the influence of different variables on the glomerular filtration rate of BKVN. The authors divided the patients into CNI reduction group, MMF reduction group, CNI shift to mTOR inhibitor group, and CNI shift to mTOR inhibitor as a second-step group. The result showed rapid viral load reduction has a significant association with stable or increasing GFR, regardless of which kind of reduction strategies (*p* = 0.0004, Log-rank test) [158]. This result was also compatible with the aforementioned studies [148,149].

Due to the lack of direct markers for renal transplant recipient immunity, the remaining option is to adjust the IS dosage by evaluating indirect assessments such as creatinine, urine, blood BKPyV DNA PCR, and donor-specific antibodies individually. Torque Teno virus (TTV), a nonpathogenic and ubiquitous virus, has gained attention to be a potential marker of immune function in solid organ transplantation [159,160]. The replication and clearance of commensal TTV viral load were under close control of our immune system [161]. IS after transplant impaired the balance, and TTV was found to increase after excessive immunosuppression [159,162]. TTV viral load also increases when patients are co-infected with other pathogens and in patients who have autoimmune inflammatory diseases [163,164,165,166,167]. This correlation provides a rationale for TTV being a promising marker of the net immunosuppression state. Schiemann et al. demonstrated that lower TTV viral load was independently associated with antibody-mediated rejection [168]. Current prospective studies revealed the value of TTV quantification for risk stratification of kidney graft rejection or infection. It might be used as a monitoring tool but not a diagnostic tool yet [169,170,171]. However, a study claimed that there is no relationship between BKPyV replication and TTV viral load [172]. Further prospective studies are warranted to confirm the clinical values of TTV quantification and its clinical use, including optimal TTV range, international unity, and hard clinical outcome prediction.

By measuring virus-specific T cell levels in pediatric post-transplant care, steering IS was presented in the IVIST trial results recently. A multicenter, randomized, controlled trial enrolled 64 pediatric KTRs. They monitored trough level in both groups and virus-specific T cell levels in the intervention group for IS dosage adjustment [173]. Compared to control groups, both everolimus and cyclosporine’s dosage was reduced in the intervention group with no difference in renal function 2 years after transplantation. Both trough levels of everolimus and cyclosporine were significantly lowered. Besides, patients in the intervention group were more likely to be spared from glucocorticoid use at 2-year post-transplant.

Meanwhile, fewer acute rejection events, similar de novo donor-specific antibody development, viral infection (CMV, *herpes simplex virus*, *Epstein-Barr virus* (EBV)), and BKVN were noted in the intervention group [173]. This study provides a safe measurement other than the pharmacokinetic method for personalizing dosing and IS reduction. That means we can avoid CNI toxicity or the side effect of long-term steroid use. Future larger trials focusing on prevention overimmunosuppression for adult transplant recipients with a standard triple regimen consisting of tacrolimus, mycophenolate mofetil, and steroid are expected. The IVIST trial may be a paradigm shift for immunoassay-guided optimal immunosuppression in future clinical practice [173].

## 6. Novel Treatment for BKVN

### 6.1. Immune Therapy

#### 6.1.1. Intravenous Immunoglobulin

The therapeutic mechanisms of intravenous immunoglobulin (IVIG) for BKVN are not fully understood. Both donated and commercial IVIG contains IgG against various infectious diseases, including BKPyV neutralizing antibodies [174,175]. Meanwhile, IVIG has powerful indirect immunomodulatory effects [176,177]. Successful case series of viremia-lowering adjunctive therapy with IVIG had been reported after the failure of IS dose reduction and leflunomide administration [178,179,180]. An additional IVIG group presented cleared viremia and BKPyV immunohistochemistry evident from repeated tissue sampling [181]. A recent study showed significant increasing BKPyV genotype-specific neutralizing antibody titers in KTRs [182]. A retrospective study showed prophylactic IVIG in the early post-transplant phase was associated with a significantly lower incidence of both BKPyV viremia and BKVN in high-risk recipients [183]. Further randomized control trials are in expectancy in this field for more substantial evidence of IVIG efficacy.

On the other hand, IVIG is also the most common therapy for antibody-mediated rejection in adjunct with plasmapheresis and/or rituximab. The plasmapheresis removes the donor-specific antibodies, and IVIG exerts immunomodulatory effects on the antibodies. A meta-analysis included 21 articles of antibody-mediated rejection since 1950, showing insufficient evidence of all kinds of treatments due to each article’s small sample size [184]. Lefaucheur et al. conducted a randomized trial that compared IVIG only or IVIG combined plasmapheresis and rituximab. The high graft loss rate in IVIG alone group indicated IVIG by itself is not enough to prevent antibody-mediated rejection. Due to limited data and sample size of studies in this field, current management for antibody-mediated rejection remains plasmapheresis and IVIG combination therapy [185].

#### 6.1.2. Cellular Therapy

The importance of cellular immunity toward BKPyV infection in transplant recipients has been recognized [186]. The BKPyV-specific T cell has drawn much attention, and its amount has a positive association with clearing BKPyV viremia in KTRs [30,187]. Failure of BKPyV-specific T cell to control viral replication due to IS overdose results in reactivation of BKPyV infection [188]. Thus, cellular therapy to regain immunity in recipients is a developing field in BKPyV immunotherapy. Owing to the advances in immunological techniques, adoptive T cell therapy was assisted by synthetic viral peptides to identify BKPyV and MHC antigens. Also, T cell expansion was performed by overlapping peptide pools. The enzyme-linked immunospot (ELISPOT) assay and tetramer staining can measure T cell responses. Many studies aimed to recognize adoptive T cell therapy’s safety and toxicity in vitro and in vivo. Papadopoulou et al. used overlapping peptide pools to generate virus-specific T cells for the commonly detected virus, including EBV, CMV, human herpesvirus 6 in vitro.

Meanwhile, these virus-specific T cells had successfully treated different viral infections, with a 94% response rate in 8 hematopoietic stem cell transplant (HSCT) patients without toxicity [189]. A phase II clinical trial showed that administration of BKPyV-specific T cells manufactured from a patient’s stem cell donor or unrelated donors could reduce symptomatic infection and BK viral load effectively in HSCT and solid organ transplant (SOT) recipients. A study enrolled 38 HSCT recipients and 3 SOT recipients who developed BKPyV viremia and/or hemorrhagic cystitis or nephropathy after transplant. The results showed clinical benefits; the overall response rate was 86% in the BK viremia group and 100% in the hemorrhagic cystitis group; 87% of patients in both groups were free of adverse effects, notably without a reduction in IS dose. This study supports further investigation in T cell therapy or even prophylaxis for BKVN [190].

### 6.2. Vaccine

There is no BKPyV vaccine currently, with most in the concept and design phase. Augmenting the humoral or cellular immune response to BKPyV is the central concept [191]. Due to cross-reaction did not exist between BKPyV serotypes, viral capsid protein aggregates instead of viral genetic components are the current approach in vaccine development [192,193]. Immunodominant peptides-modified BKPyV has been investigated [194]. Recent research found the multi-epitope vaccine with potential effectiveness may solve problems mention above for wide population use. Although the results are still in the experiment phase, it still displays impressive advances in this field [195].

## 7. Conclusions

BKPyV has a significant impact on kidney allograft during the first year post-transplant. Measures including preemptive monitoring combined with timely IS dose reduction decrease the graft failure rate caused by BKVN. The optimal IS regimen is to balance rejection and infection through delicate clinical evaluations (Figure 3). Meanwhile, evidence suggests that an mTOR inhibitor-based regimen may be beneficial to treat BKVN. Understanding the pre-and post-transplant risk factors helps us reduce complications. The step-by-step nature of international standard guidelines for tailoring IS dosage has its limitation due to ethical dilemmas, subsequent rejection, and graft injury. Direct markers for immunity assessment and direct antiviral agents are further research objectives. With the advances in immunology and biological sciences, we can look forward to a new era in the diagnosis and therapy of BKVN.

## Figures and Tables

**Figure 1 viruses-13-00487-f001:**
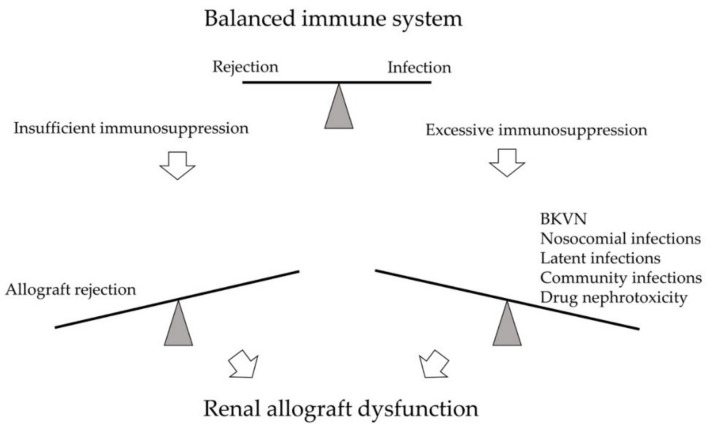
The immune system of kidney transplant recipients is balanced between rejection and infection. Excessive immunosuppression may lead to infections, such as BK polyomavirus nephropathy (BKVN), nosocomial infections, latent infections, and community infections. Drug nephrotoxicity may also develop. On the opposite side, insufficient immunosuppression may result in allograft rejection. Both arms may cause significant kidney damage and renal allograft dysfunction.

**Figure 2 viruses-13-00487-f002:**
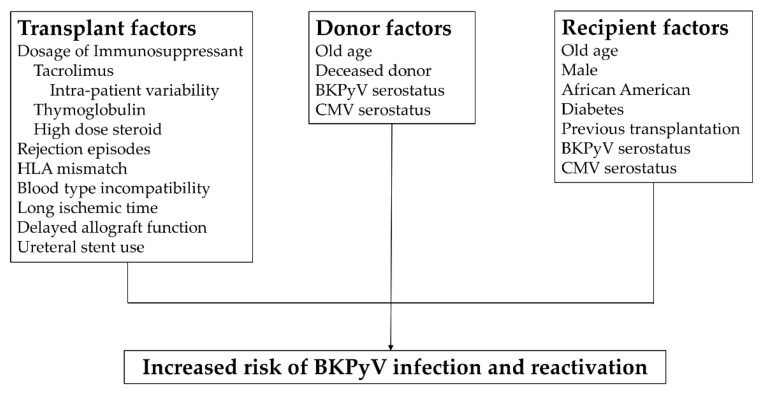
Risk factors for BKPyV infection. Risk factors can be assorted into 3 categories: Transplant factors [39,56,62,63,64,65,67,68,70,71,72], donor factors [16,18,62,64,66,69], and recipient factors [16,18,56,64,69]. Understanding risk factors that affect before and after transplant can be helpful in immune balance. Abbreviations: HLA, human leukocyte antigen; BKPyV, BK polyomavirus; CMV, cytomegalovirus.

**Figure 3 viruses-13-00487-f003:**
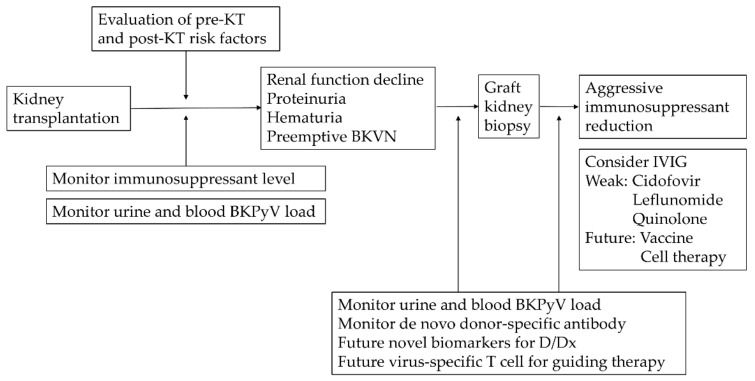
Conceptual illustration of evaluations, screening, diagnosis, and management for BKPyV infection. Abbreviations: KT, kidney transplant; BKPyV, BK polyomavirus; BKVN, BK polyomavirus nephropathy; IVIG, intravenous immunoglobulin; D/Dx, differential diagnosis.

## Data Availability

The data that support the findings of this study are available from the corresponding author upon reasonable request.

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
