# Peer review of "BK Polyomavirus Nephropathy in Kidney Transplantation: Balancing Rejection and Infection"

_viruses, 2021, doi:10.3390/v13030487_

Round 1

Reviewer 1 Report

1. Abbreviation is BKVAN instead of BKVN

2. Abstract line 37: should be more clear to not imply that ‘’reactivated’’ also concerns the virus from the donor. Use ‘’infection’’ instead of ‘’reactivation’’ for the virus from the donor.

3. Line 39: Not clear, revise the sentence’s structure

4. Line 43: replace ‘’How to keep’’ with ‘’keeping’’

5. Line 58: seroprevalence of BKV means that of all serotypes. Most studies that mention an 80 to 90% seroprevalence only take the Ib2 serotype into consideration.

6. Line 62 ‘’After the primary infection, the virus-cell remains latent in the kidney…’’: The virus remains latent not the cell.

7. Line 159: BV

8. Major English language review

9. List of abbreviations? 

10. Line 205 you can add reference for a recent review: doi:10.3390/v11100945

11. Studies in references 64 and 67 included 173 and 168 KTRs respectively. Study with reference 63 included 203 KTRs. It is also considered ‘’ smaller scale study’’ compared to that of Wunderink et al (407 donor-recipient pairs).

12. Part 6, Treatment: A recent review (PMID: 30958614) has already presented in details the various choices for BKV treatment: there is less need to review treatments and more need to focus on the balance between the need for immunosuppression and the BKV infection risk (as the title implies). Discuss more about the anti-BKV immune response.

13. Important references

  1. ^ Gupta G, Shapiro R, Thai N, Randhawa PS, Vats A (August 2006). "Low incidence of BK virus nephropathy after simultaneous kidney pancreas transplantation". Transplantation82 (3): 382–8. doi:10.1097/01.tp.0000228899.05501.a7. PMID 16906037. S2CID 12310204.
  2. ^ Egli A; Infanti L; Dumoulin A; et al. (2009). "Prevalence of polyomavirus BK and JC infection and replication in 400 healthy blood donors". J Infect Dis199 (6): 837–46. doi:10.1086/597126. PMID 19434930.
  3. ^ Fishman, J. A. (2002). "BK Virus Nephropathy — Polyomavirus Adding Insult to Injury". New England Journal of Medicine347 (7): 527–530. doi:10.1056/NEJMe020076. PMID 12181409.
  4. ^ Elfadawy, NS; Flechner, SM; Xiaobo, L; Schold, J; Tian, D; Srinivas, TR; Poggio, E; Fatica, R; Avery, R; Mosaad, SB (2013). "The impact of surveillance and rapid reduction in immunosuppression to control BK virus-related graft injury in kidney transplantation". Transplant International26 (8): 822–32. doi:10.1111/tri.12134. PMID 23763289.

Author Response

Reviewer 1:

  1. Abbreviation is BKVAN instead of BKVN

=> Responses: We truly appreciate this valuable comment. We performed the search ‘BKVN’ and ‘BKVAN’ on the PubMed page. Both BKVAN (116 results; https://pubmed.ncbi.nlm.nih.gov/?term=bkvan) and BKVN (151 results; https://pubmed.ncbi.nlm.nih.gov/?term=bkvn) are commonly used in the literature. However, it seems that more authors used BKVN instead of using BKVAN in their articles. Thus, we are considering keeping the abbreviation BKVN.

  1. Abstract line 37: should be more clear to not imply that ‘’reactivated’’ also concerns the virus from the donor. Use ‘’infection’’ instead of ‘’reactivation’’ for the virus from the donor.

=> Responses: Thank you for your valuable comment. We have revised this sentence on page 1, lines 37-39, as follows. The leading cause of BKVN is overimmunosuppression that reactivated the latent BK virus (BKV) within the recipient or reinforced BKV infection from the allograft.

  1. Line 39: Not clear, revise the sentence’s structure.

=> Responses: Thank you for your valuable comment. The sentence was revised on page 1, lines 40-42, as follows. No effective direct antiviral therapy is currently available; thus, since the first case was identified in 1971, immunosuppressant (IS) reduction remains the primary strategy for BKVN.

  1. Line 43: replace ‘’How to keep’’ with ‘’keeping.’’

=> Responses: Thank you for your valuable comment. The sentence was revised in the introduction section on page 2, line 47, as follows. Therefore, the highest principle in clinical practice is keeping a balance between rejection and infection.

  1. Line 58: seroprevalence of BKV means that of all serotypes. Most studies that mention an 80 to 90% seroprevalence only take the Ib2 serotype into consideration.

=> Responses: We truly appreciate your valuable comment. We have revised the sentence and the cited articles in the “2.About the BKV” section on page 3, lines 102-103, as follows: During the first decade of childhood, the primary exposure, often with subclinical symptoms, resulted in 80-90% of adults developed antibodies against BKV (Stolt, Sasnauskas, et al. 2003) (Egli, Infanti, et al. 2009).

  1. Line 62 ‘’After the primary infection, the virus-cell remains latent in the kidney...’’: The virus remains latent not the cell.

=> Responses: Thank you for your valuable comment. The word “cell” has been deleted in the “2.About the BKV” section on page 3, line 106, as follows. After the primary infection, the virus remains latent in the kidney, peripheral-blood leucocytes, and possibly the brain.

  1. Line 159: BV

=> Responses: Thank you for your valuable comment. We have replaced “BKV” with the typo “BV” on page 5, line 191.

  1. Major English language review

=> Responses: We truly appreciate your valuable comment. A native English speaker who is a medical expert did the English editing throughout the whole manuscript.

  1. List of abbreviations?

=> Responses: Thank you for your valuable comment. According to the author's instructions of this journal, abbreviations should be defined in parentheses the first time they appear in the abstract, main text, and in figure or table captions and used consistently thereafter. Therefore, we are considering not presenting the list of abbreviations.

  1. Line 205 you can add reference for a recent review: doi:10.3390/v11100945.

=> Responses: We truly appreciate your valuable comment. We cited this review article as reference 94 in Section 4.1, “Risk factors for BKV infection or reactivation,” on page 6, line 240 (Dakroub, Touze, et al. 2019).

  1. Studies in references 64 and 67 included 173 and 168 KTRs, respectively. Study with reference 63 included 203 KTRs. It is also considered “smaller scale study” compared to that of Wunderink et al. (407 donor-recipient pairs).

=> Responses: Thank you for your valuable comment. Due to the latter studies are not smaller than previous studies that we cannot use “smaller” for description. We have revised this sentence in Section 4.1, “Risk factors for BK virus infection or reactivation,” on page 6, line 239, as follows. Several “other” studies reached the same conclusion through ELISA or neutralization inhibition assay.

  1. Part 6, Treatment: A recent review (PMID: 30958614) has already presented in details the various choices for BKV treatment: there is less need to review treatments and more need to focus on the balance between the need for immunosuppression and the BKV infection risk (as the title implies). Discuss more about the anti-BKV immune response.

=> Responses: We truly appreciate your valuable comment. We have condensed and merged parts 6.1 and 6.2 into one concise paragraph, located on page 12, lines 525-539, and page 13, lines 540-545.

Besides, we also added the relevant content of anti-BKV immune response in Section 2, “About the BKV,” on page 2, lines 76-78, and page 3, lines 79-101, as follows. Cellular immunity is the most determining part of the immune response for controlling BKV viremia and BKVN. Innate immune response has its important role in the primary infection (Comoli, Binggeli, et al. 2006). Besides, dendritic cells are critical in the induction of adaptive immune response (Drake, Moser, et al. 2000). Womer et al. reported lower numbers of dendritic cells in peripheral blood were noted in those KTRs developed BKVN. They also revealed KTRs with fewer dendritic cells before transplants are more associated with BKVN (Womer, Huang, et al. 2010). Furthermore, BKV can decrease the natural killer cell-mediated cytotoxicity by inhibiting the identification of natural killer cells (Bauman, Nachmani, et al. 2011). Other innate immune mediators are associated with renal inflammation(Ribeiro, Wornle, et al. 2012). Adaptive immune response develops after exposure to viral antigens. Humoral response works via neutralizing antibodies to defend the further viral infectious process. Studies showed seronegative recipients have higher risks in viremia and BKVN than seropositive recipients that humoral immunity may help limit BKV infection (Ginevri, De Santis et al. 2003, Smith, McDonald, et al. 2004, Bohl, Storch, et al. 2005, Sood, Senanayake, et al. 2013, Abend, Changala, et al. 2017, Wunderink, van der Meijden, et al. 2017). Meanwhile, recipients paired with seropositive donors have higher post-transplant BK-specific-antibody titer than the seronegative donor group (Andrews, Shah, et al. 1988). It means that BKV infection from the donor can induce the humoral immune response (Lamarche, Orio, et al. 2016). However, the virus can hide away from neutralization with a mutation in viral antibody receptors (Ciarlet, Hoshino, et al. 1997, Li, Diprimio, et al. 2012). In this situation, latent viral reactivation can be well-controlled by antiviral memory T cells (Ambalathingal, Francis, et al., 2017). Cellular immunity can help for more effective infection control because of detection and cytotoxicity (Lamarche, Orio, et al. 2016). Both CD4+ and CD8+ T cells are important, especially the polyfunctional BKV-specific T cells (Renner, Dietrich, et al. 2013, Schaenman, Korin, et al., 2017). After kidney transplant, KTRs with viruria but no progression to viremia have positive BKV-specific T cell response (Comoli, Azzi, et al. 2004). Conversely, there is no BKV-specific T cell response in KTRs with BKV viremia or BKVN (Comoli, Azzi et al. 2004, Comoli, Hirsch et al. 2008, Schachtner, Muller et al. 2011, Schachtner, Stein et al. 2014, Schachtner, Stein et al. 2015). Also, quick BK-specific T cell response was noted in the viremia-resolved group, while the response was only reported after reduced IS in the developed BKVN group (Ozsancak, Auzou, et al. 2004, Schachtner, Muller, et al. 2011). Studies concluded that it is crucial for KTRs to reconstitute the BKV-specific T cells when BKV infection.

Important references:

  • Gupta G, Shapiro R, Thai N, Randhawa PS, Vats A (August 2006). "Low incidence of BK virus nephropathy after simultaneous kidney-pancreas transplantation." Transplantation. 82 (3): 382–8. doi:10.1097/01.tp.0000228899.05501.a7. PMID 16906037. S2CID 12310204.

=> Responses: We truly appreciate your valuable comment. Gupta G et al. reported their experience of a lower incidence of BKVN in simultaneous kidney-pancreas transplantation recipients by receiving modified immunosuppression with antibody preconditioning at the University of Pittsburgh Medical Center. They concluded the reasons, including less toxic immunosuppressive protocols, earlier diagnosis, and the use of antiviral therapy. In our articles, we mentioned close monitoring of clinical markers and the strategies for immunosuppression reductions. Protocols of immunosuppressants and antiviral agents lack international guidelines, and successful single-center studies provide physicians with clinical consultation. We have cited this important article as reference 160 in the sentence in Section 5, “Strategies of immunosuppression reduction,” on page 10, lines 453-454, as follows. Accounting for the lack of publishment in addition to the different responses of individuals toward IS, further management should be based on expert opinion in each clinical situation [160].

  • Egli A; Infanti L; Dumoulin A; et al. (2009). "Prevalence of polyomavirus BK and JC infection and replication in 400 healthy blood donors". J Infect Dis. 199 (6): 837–46. doi:10.1086/597126. PMID 19434930.

=> Responses: We truly appreciate your valuable comment. Egli A et al. presents an important epidemiological study about the prevalence in immunocompetent individuals. We have cited this article as reference 33 in the sentence in Section 2, “About the BKV,” on page 3, lines 102-103, as follows. During the first decade of childhood, the primary exposure, often with subclinical symptoms, resulted in 80-90% of adults developed antibodies against BKV (Stolt, Sasnauskas, et al. 2003) (Egli, Infanti, et al. 2009).

  • Fishman, J. A. (2002). "BK Virus Nephropathy — Polyomavirus Adding Insult to Injury." New England Journal of Medicine. 347 (7): 527–530. doi:10.1056/NEJMe020076. PMID 12181409.

=> Responses: We truly appreciate your valuable comment. Fishman et al. discussed the rationale of BKVN prevention and the flow of management after BKV infection. We have cited this important article as reference 3in the introduction section on page 2, line 47, as follows. Therefore, the highest principle in clinical practice is keeping a balance between rejection and infection [3].

  • Elfadawy, NS; Flechner, SM; Xiaobo, L; Schold, J; Tian, D; Srinivas, TR; Poggio, E; Fatica, R; Avery, R; Mosaad, SB (2013). "The impact of surveillance and rapid reduction in immunosuppression to control BK virus-related graft injury in kidney transplantation." Transplant International. 26 (8): 822– 32. doi:10.1111/tri.12134. PMID 23763289.

=> Responses: We truly appreciate your valuable comment. Elfadawy NS et al. reported early surveillance for BK viremia to rapid reduction of immunosuppression limited the incidence of BKVAN to 1.3%. This article supports our manuscript emphasizing the importance of close monitoring for BK viral load. We have cited this article as reference 159 in the sentence in Section 5 on page 10, lines 449-450, as follows. Donor-specific antibody and blood viral load PCR can be followed up for evaluations.

Reviewer 2 Report

Revision manuscript viruses-1113119 entitled” BK Virus Nephropathy in Kidney Transplantation: Balancing Rejection and Infection” by Shen et al.

The review addresses the well-known problem of balancing the immunosuppression to avoid allograft rejection and controlling the BKPyV reactivation. The review is potentially very interesting and presents a lot of information about the topic, but it is very difficult to read and probably too long. I would suggest to revising extensively the manuscript for the English language and the organization.

Minor points:

-Use BKPyV instead of BKV

Author Response

Reviewer 2: The review addresses the well-known problem of balancing the immunosuppression to avoid allograft rejection and controlling the BKV reactivation. The review is potentially very interesting and presents a lot of information about the topic, but it is very difficult to read and probably too long. I would suggest revising extensively the manuscript for the English language and the organization.

  1. Minor points: Use BKV instead of BKV

=> Responses: We genuinely appreciate your comment. We have already performed the PubMed search “BKV” (303 results; https://pubmed.ncbi.nlm.nih.gov/?term=BKV) and “BKV” (1,564 results; https://pubmed.ncbi.nlm.nih.gov/?term=BKv). Both terms are commonly used in the literature, but it seems that more authors used BKV instead of using BKV in their articles. Thus, we are considering keeping BKV as the abbreviation of the BK virus.

Reviewer 3 Report

Chia-Lin Shen and colleagues in this manuscript revised the literature concerning BKPyV infection and associated Nephropathy in kidney transplantation. In the article the authors discussed the balancing between rejection and BKPyV infection reporting old and new therapeutic intervention, laboratory and immunological aspects for the diagnosis of BKPyV associated diseases. Moreover, it was described the new biomarkers investigation for the assessment of immunosuppression and BKPyV reactivation and development of Nephropathy. The authors have done an extensive revision of the argument and produced a near exhaustive and high-quality work. To our thinking, there are few points that could be relevant to complete the review.  

Main points
1- Page 2, section 2: The authors should add a brief description of the BKPyV genome, reporting the early and late expression regions with the principal expressed proteins, the non-coding control region (NCCR) and the potential microRNA expression. Moreover, a brief comment on the role of NCCR in the viral reactivation should be also included.
2- page 10, lines 350-354: the authors reported a study of the relationship of BKPyV microRNA expression and BKPyV associated disease. In this context, there are other previous studies that approached this issue that should be cited. 
3- page 13, paragraph starting with “Due to the lack…: In the last years several investigations regarding additional marker to be monitored for immunosuppressed status in transplanted patients are emerged. Thus there are data reporting the correlation of TTV (Torque teno virus a component of human virome) and BKV viral load that seems to be an indicator for the immune status of the host. Comment about these with the recent data could be relevant in this review. 

Minor points
1- page 4, line 147: the term “disease” should be changed with “infection”.
2- page 5, line 159: “BV” should be changed with “BK”.
3- page 5, line 157: “with” it should be removed.
4- page 5, figure 2: the reference included in the figure should be numerically progressive. Thus the reference “103” should be changed.
5- page 6, line 198 and page 13, line 454: the statistic method used should be reported (chi square? student’t test ?). 
6- page 7, line 244: “African American” should be correct with “African-American”.
7-  page 11, lines 384-386: the phrases should be corrected. 

Author Response

Reviewer 3: Chia-Lin Shen and colleagues in this manuscript revised the literature concerning BKV infection and associated Nephropathy in kidney transplantation. In the article, the authors discussed the balancing between rejection and BKV infection reporting old and new therapeutic intervention, laboratory and immunological aspects for the diagnosis of BKV associated diseases. Moreover, it was described the new biomarkers investigation for the assessment of immunosuppression and BKV reactivation and development of Nephropathy. The authors have done an extensive revision of the argument and produced a near exhaustive and high-quality work. To our thinking, there are few points that could be relevant to complete the review.

Main points:

  1. Page 2, section 2: The authors should add a brief description of the BKV genome, reporting the early and late expression regions with the principal expressed proteins, the non-coding correlation control region (NCCR) and the potential microRNA expression. Moreover, a brief comment on the role of NCCR in the viral reactivation should be also included.

=> Responses: We truly appreciate this valuable comment. We have added the relevant content in Section 2, “About the BKV,” on page 2, lines 63-75, as follows. BKV is a double-stranded DNA virus, and its genome consists of the early coding region, late coding region, and a non-coding control region (NCCR) in between (Moens, Van Ghelue, et al. 2007). The early region usually codes for the replication proteins, the small tumor antigens, the large tumor antigens (TAgs), and agnoprotein. The late region codes for expressing structural proteins VP1, VP2, and VP3 (Helle, Brochot, et al. 2017). The microRNAs expression was transcribed from the 3’ end of the TAgs and act as a regulator in BKV infection (Broekema and Imperiale 2013). The NCCR contains the genome of promotors of the early and late region, transcriptional start sites, and the origin of replication. It also provides binding sites for host cellular regulatory factors. NCCR variation exists between BKV isolates, and the rearranged forms of NCCR are associated with disease (Cubitt 2006). Studies concluded that the high heterogeneity of NCCR is considered the ability of environmental adaptation and higher pathogenicity for disease progression (Helle, Brochot, et al. 2017).

  1. page 10, lines 350-354: the authors reported a study of the relationship of BKV microRNA expression and BKV associated disease. In this context, there are other previous studies that approached this issue that should be cited.

=> Responses: We truly appreciate this valuable comment. After a literature review, we found two studies that are relevant to this issue. Tian et al. reported miR-B1 expression was upregulated during BKV infection (Tian, Li et al. 2014). Li et al. found a relationship between levels of bkv-miR-B1-5p and the presence of biopsy-proven BK viral nephropathy (Li, McNicholas, et al. 2014). Both studies used BKV microRNA as early diagnostic tools. We have added these two articles as references 146 and 147 in Section 4.4, “Biologic marker development in BKVN,” on page 9, lines 383-386, as follows. It has been reported that urinary exosomal BK viral microRNA, bkv-miR-B1-5p and bkv-miR-B1-5p/miR-16, have excellent statistical significance for the diagnosis of BKVN, with the area under the curve values of 0.989 and 0.985, respectively.

  1. page 13, paragraph starting with “Due to the lack...: In the last years several investigations regarding additional marker to be monitored for immunosuppressed status in transplanted patients are emerged. Thus, there are data reporting the correlation of TTV (Torque teno virus a component of human virome) and BKV viral load that seems to be an indicator for the immune status of the host. Comment about these with the recent data could be relevant in this review.

=> Responses: We truly appreciate this valuable comment. We have added the relevant content in Section 5, “Strategies of immunosuppression reduction,” on page 12, lines 491-507, as follows. Torque Teno virus (TTV), a nonpathogenic and ubiquitous virus, has gained attention to be a potential marker of immune function in solid organ transplantation (De Vlaminck, Khush, et al., 2013, Focosi, Macera, et al. 2014). The replication and clearance of commensal TTV viral load were under close control of our immune system (Maggi, Pistello, et al. 2001). Immunosuppressants after transplant impaired the balance that TTV was found to increase in excessive immunosuppression (Focosi, Macera, et al. 2014, Strassl, Schiemann, et al. 2018). TTV viral load is also higher when patients are co-infected with other pathogens and in patients with autoimmune of inflammatory diseases (Focosi, Maggi, et al. 2010, Wootton, Kim, et al. 2011, Borkosky, Whitley, et al. 2012, Haloschan, Bettesch, et al. 2014, Rigante, Mazzoni, et al. 2014). This correlation provides the rationale for TTV being a potential monitor marker for net immunosuppression state. Schiemann et al. demonstrated that lower TTV viral load was independently associated with antibody-mediated rejection (Schiemann, Puchhammer-Stockl, et al. 2017). Current prospective studies about TTV and kidney transplants showed evidence for the values of TTV quantification for risk stratification of graft rejection and infection as a monitor tool, not a diagnostic tool yet (Fernandez-Ruiz, Albert, et al. 2019, Solis, Velay, et al. 2019, Doberer, Schiemann, et al. 2020). However, there is no relationship between BKV replication and TTV viral load (Handala, Descamps, et al., 2019). Further prospective studies may evaluate TTV as a clinical monitor tool for KTR immunity, such as optimal TTV range, international unity, and hard clinical outcome.

Minor points:

  1. Page 4, line 147: the term “disease” should be changed to “infection.”

=> Responses: Thank you for your valuable comment. We have revised the sentence in Section 4, “Balance the rejection and infection clinically,” on page 5, line 181, as follows. The primary infection of BKV usually occurs during childhood.

  1. Page 5, line 159: “BV” should be changed with “BK.”

=> Responses: Thank you for your valuable comment. We have replaced “BKV” with the typo “BV” on page 5, line 191.

  1. Page 5, line 157: “with” it should be removed.

=> Responses: Thank you for your valuable comment. We have removed the word “with” on page 5, lines 205-206, as follows. Prince et al. suggested that BKVN only manifests while the host immunity is over suppressed, whereas acute rejection independently plays a role regardless of therapeutic regimens.

  1. Page 5, figure 2: the reference included in the figure should be numerically progressive. Thus the reference “103” should be changed.

=> Responses: We truly appreciate your comment. We have corrected the references in Figure 2 on page 5.

  1. Page 6, line 198 and page 13, line 454: the statistic method used should be reported (chi square? Student's test?).

=> Responses: We truly appreciate your comment. For page 6, line 198, Wunderink et al. designed this study in order to investigate whether BKV seroreactivity of the donor predicts viremia and BKVN in the recipient (Wunderink, van der Meijden et al. 2017). They enrolled the data of donor BKV seropositivity and seroreactivity. Also, they divided BKV seroreactivity into quartile groups. The recipients were separated to viremic and non-viremia groups. In addition, the viremic recipients were divided as BKVN group or no-BKVN group. The authors compared these factors by the Student’s t-test. Thus, we revised the sentence in Section 4.1, “Risk factors for BK virus infection or reactivation,” on page 6, lines 232-233, as follows. Wunderink et al. published the largest research until now that revealed donor seropositivity was strongly associated with the occurrence of recipient viremia and BKVN (p < 0.001, Student’s t-test). For page 13, line 454, Schwarz et al. retrospectively studied the influence of different variables on the glomerular filtration rate of BKVN(Schwarz, Linnenweber-Held et al. 2012). The authors divided the patients into CNI reduction group, MMF reduction group, CNI shift to mTOR group, and CNI shift to mTOR as a second step group. They composed survival analysis by Cox regression analysis and followed by Log-rank test. Thus, we revised the sentence in Section 5, “Strategies of immunosuppression reduction,” on page 11, line 487, as follows. The result showed rapid viral load reduction has a significant association with stable or increasing GFR, regardless of which kind of reduction strategies (p = 0.0004, Log-Rank test).

  1. Page 7, line 244: “African American” should be correct with “African-American.”

=> Responses: Thank you for your valuable comment. We have corrected this term and on page 7, line 276, and page 9, line 372.

  1. Page 11, lines 384-386: the phrases should be corrected.

=> Responses: Thank you for your valuable comment. We have added the phrase “when” at the beginning of the sentence, on page 10, line 417, as follows. When a kidney transplant recipient's immune system is over-suppressed, the incidence of opportunistic infection from BKV, CMV, John Cunningham virus, Pneumocystis jiroveci, and other fungi increase.

Round 2

Reviewer 2 Report

-Use BKPyV instead of BKV as indicated by the new nomenclature (Archives of Virology volume 161, pages1739–1750(2016)

“promotors” replace with “promoters”

-“Womer et al. reported a lower number of dendritic cells in pe-ripheral blood was noted in those KTRs developed BKVN”, revise the sentence.

-There are still some grammar mistakes

As I said previously I would have prefered a shorter review.

Author Response

Point-by-point responses to editor and reviewers (viruses-1113119.R2)

Reviewer 2:

1. Use BKPyV instead of BKV as indicated by the new nomenclature (Archives of virology volume 161, pages 1739-1750(2016).

=> Responses: We truly appreciate this valuable comment. We have replaced BKV with BKPyV in the whole article per your suggestion.

2. “promotors” replace with “promoters.”

=> Responses: Thank you for your valuable comment. We have replaced “promotors” with “promoters” on page 2, lines 66-67, as follows. The NCCR contains the genome of promoters of the 66 early and late regions, transcriptional start sites, and the origin of replication.

3. Womer et al. reported a lower number of dendritic cells in peripheral blood was noted in those KTRs developed BKVN,” revise the sentence.

=> Responses: We truly appreciate this comment. The sentence has been revised in the “2. About the BKPyV” section on page 2, lines 75-77, as follows. Womer et al. reported that the number of peripheral blood dendritic cells is lower in KTRs who developed BKVN.

4. There are still some grammar mistakes.

=> Responses: We truly appreciate your valuable comment. A native English speaker who is a medical expert has done the English editing and grammar check throughout the whole manuscript during this revision.

5. As I said previously I would have preferred a shorter review.

=> Responses: Thank you for your valuable comment. We have deleted the “6.1 Cidofovir, leflunomide and fluoroquinolones” section and the “6.4 Retransplantation” section. We also condensed other sections, and the deleted contents were marked in red color.